# Protein translocation channel of mitochondrial inner membrane and matrix-exposed import motor communicate via two-domain coupling protein

Rupa Banerjee[1], Christina Gladkova[1], Koyeli Mapa[1], Gregor Witte[2], Dejana Mokranjac[1]*

[1]Biomedical Center Munich, Department of Physiological Chemistry, Ludwig-Maximilians-Universität, Munich, Germany; [2]Gene Center, Department of Biochemistry, Ludwig-Maximilians-Universität, Munich, Germany

**Abstract** The majority of mitochondrial proteins are targeted to mitochondria by N-terminal presequences and use the TIM23 complex for their translocation across the mitochondrial inner membrane. During import, translocation through the channel in the inner membrane is coupled to the ATP-dependent action of an Hsp70-based import motor at the matrix face. How these two processes are coordinated remained unclear. We show here that the two domain structure of Tim44 plays a central role in this process. The N-terminal domain of Tim44 interacts with the components of the import motor, whereas its C-terminal domain interacts with the translocation channel and is in contact with translocating proteins. Our data suggest that the translocation channel and the import motor of the TIM23 complex communicate through rearrangements of the two domains of Tim44 that are stimulated by translocating proteins.

*For correspondence: dejana.
mokranjac@med.uni-muenchen.
de

**Competing interests:** The authors declare that no competing interests exist.

## Introduction

Mitochondria perform a number of essential cellular functions ranging from production of ATP and diverse other metabolic intermediates to initiation of apoptosis. It is thus not very surprising that disturbances in mitochondrial function are associated with a number of human diseases, including neurodegenerative disorders, diabetes, and various forms of cancer (*Nunnari and Suomalainen, 2012*; *Quirós et al., 2015*; *Youle and van der Bliek, 2012*). An essential prerequisite for correctly functioning mitochondria is import of about 1000 different proteins synthesized as precursor proteins in the cytosol. Recent studies revealed that mitochondrial protein import machineries are sensitive indicators of functionality of mitochondria (*Harbauer et al., 2014*; *Nargund et al., 2012*; *Yano et al., 2014*), demonstrating that a deep understanding of mitochondrial protein import pathways and their regulation will be essential for understanding the role mitochondria have under physiological and pathophysiological conditions. Over half of mitochondrial proteins are synthesized with cleavable, N-terminal extensions called presequences. Import of such precursor proteins requires a coordinated action of the TOM complex in the outer membrane and the TIM23 complex in the inner membrane and is driven by membrane potential across the inner membrane and ATP in the matrix (*Dolezal et al., 2006*; *Endo et al., 2011*; *Koehler, 2004*; *Mokranjac and Neupert, 2009*; *Neupert and Herrmann, 2007*; *Schulz et al., 2015*; *Stojanovski et al., 2012*).

**eLife digest** Human, yeast and other eukaryotic cells contain compartments called mitochondria. These compartments are surrounded by two membranes and are most famous for their essential role in supplying the cell with energy. While mitochondria can make a few of their own proteins, the vast majority of mitochondrial proteins are produced elsewhere in the cell and are subsequently imported into mitochondria. During the import process, most proteins need to cross both mitochondrial membranes.

Many mitochondrial proteins are transported across the inner mitochondrial membrane by a molecular machine called the TIM23 complex. The complex forms a channel in the inner membrane and contains an import motor that drives the movement of mitochondrial proteins across the membrane. However, it is not clear how the channel and import motor are coupled together. There is some evidence that a protein within the TIM23 complex called Tim44 – which is made of two sections called the N-terminal domain and the C-terminal domain – is responsible for this coupling. It has been suggested that mainly the N-terminal domain of Tim44 is required for this role.

Banerjee et al. used biochemical techniques to study the role of Tim44 in yeast. The experiments show that both the N-terminal and C-terminal domains are essential for its role in transporting mitochondrial proteins. The N-terminal domain interacts with the import motor, whereas the C-terminal domain interacts with the channel and the mitochondrial proteins that are being moved.

Banerjee et al. propose a model of how the TIM23 complex works, in which the import of proteins into mitochondria is driven by rearrangements in the two domains of Tim44. A future challenge is to understand the nature of these rearrangements and how they are influenced by other components of the TIM23 complex.

The TIM23 complex mediates translocation of presequence-containing precursor proteins into the matrix as well as their lateral insertion into the inner membrane. The latter process requires the presence of an additional, lateral insertion signal. After initial recognition on the intermembrane space side of the inner membrane by the receptors of the TIM23 complex, Tim50 and Tim23, precursor proteins are transferred to the translocation channel in the inner membrane in a membrane-potential dependent step (*Bajaj et al., 2014*; *Lytovchenko et al., 2013*; *Mokranjac et al., 2009*; *Shiota et al., 2011*; *Tamura et al., 2009*). The translocation channel is formed by membrane-integrated segments of Tim23, together with Tim17 and possibly also Mgr2 (*Alder et al., 2008*; *Demishtein-Zohary et al., 2015*; *Ieva et al., 2014*; *Malhotra et al., 2013*). At the matrix-face of the inner membrane, precursor proteins are captured by the components of the import motor of the TIM23 complex, also referred to as PAM (presequence translocase-associated motor). Its central component is mtHsp70 whose ATP hydrolysis-driven action fuels translocation of precursor proteins into the matrix (*De Los Rios et al., 2006*; *Liu et al., 2003*; *Neupert and Brunner, 2002*; *Schulz and Rehling, 2014*). Multiple cycles of mtHsp70 binding to and release from translocating proteins are required for complete translocation across the inner membrane. The ATP hydrolysis-driven cycling of mtHsp70 and thereby its binding to proteins is regulated by the J- and J-like proteins Tim14(Pam18) and Tim16(Pam16) as well as by the nucleotide-exchange factor Mge1 (*D'Silva et al., 2003*; *Kozany et al., 2004*; *Mapa et al., 2010*; *Mokranjac et al., 2006*; *2003b*; *Truscott et al., 2003*). Tim21 and Pam17 are two nonessential components that bind to Tim17-Tim23 core of the TIM23 complex and appear to modulate its activity in a mutually antagonistic manner (*Chacinska et al., 2005*; *Popov-Celeketic et al., 2008*; *van der Laan et al., 2005*).

The translocation channel and the import motor of the TIM23 complex are thought to be coupled by Tim44, a peripheral inner membrane protein exposed to the matrix (*D'Silva et al., 2004*; *Kozany et al., 2004*; *Schulz and Rehling, 2014*). Like other components of the TIM23 complex, Tim44 is a highly evolutionary conserved protein and is encoded by an essential gene. In mammals, Tim44 has been implicated in diabetes-associated metabolic and cellular abnormalities (*Wada and Kanwar, 1998*; *Wang et al., 2015*). A novel therapeutic approach using gene delivery of Tim44 has recently shown promising results in mouse models of diabetic nephropathy (*Zhang et al., 2006*). In addition, mutations in Tim44 were identified that predispose carriers to oncocytic thyroid carcinoma

(*Bonora et al., 2006*). Understanding the function of Tim44 and its interactions within the TIM23 complex will therefore be essential for understanding how the energy of ATP hydrolysis is converted into unidirectional transport of proteins into mitochondria and may provide clues for therapeutic treatment of human diseases.

Tim44 binds to the Tim17-Tim23 core of the translocation channel (*Kozany et al., 2004*; *Mokranjac et al., 2003b*). Tim44 also binds to mtHsp70, recruiting it to the translocation channel. The interaction between Tim44 and mtHsp70 is regulated both by nucleotides bound to mtHsp70 as well as by translocating proteins (*D'Silva et al., 2004*; *Liu et al., 2003*; *Slutsky-Leiderman et al., 2007*). Tim44 is likewise the major site of recruitment of the Tim14-Tim16 subcomplex, recruiting them both to the translocation channel as well as to mtHsp70 (*Kozany et al., 2004*; *Mokranjac et al., 2003b*). In this way, Tim44 likely ensures that binding of mtHsp70 to the translocating polypeptides, regulated by the action of Tim14 and Tim16, takes place right at the outlet of the translocation channel in the inner membrane.

Tim44 is composed of two domains, depicted as N- and C-terminal domains (*Figure 1A*). Recent studies suggested that the N-terminal domain is responsible for the majority of known functions of Tim44. Segments of the N-terminal domain were identified that are important for interaction of Tim44 with Tim16 and with mtHsp70 (*Schilke et al., 2012*; *Schiller et al., 2008*). Furthermore, using site-specific crosslinking, residues in the N-terminal domain were crosslinked to the matrix-exposed loop of Tim23 (*Ting et al., 2014*). However, the C-terminal domain of Tim44 shows higher evolutionary conservation. Still, the only function that has so far been attributed to the C-terminal domain is

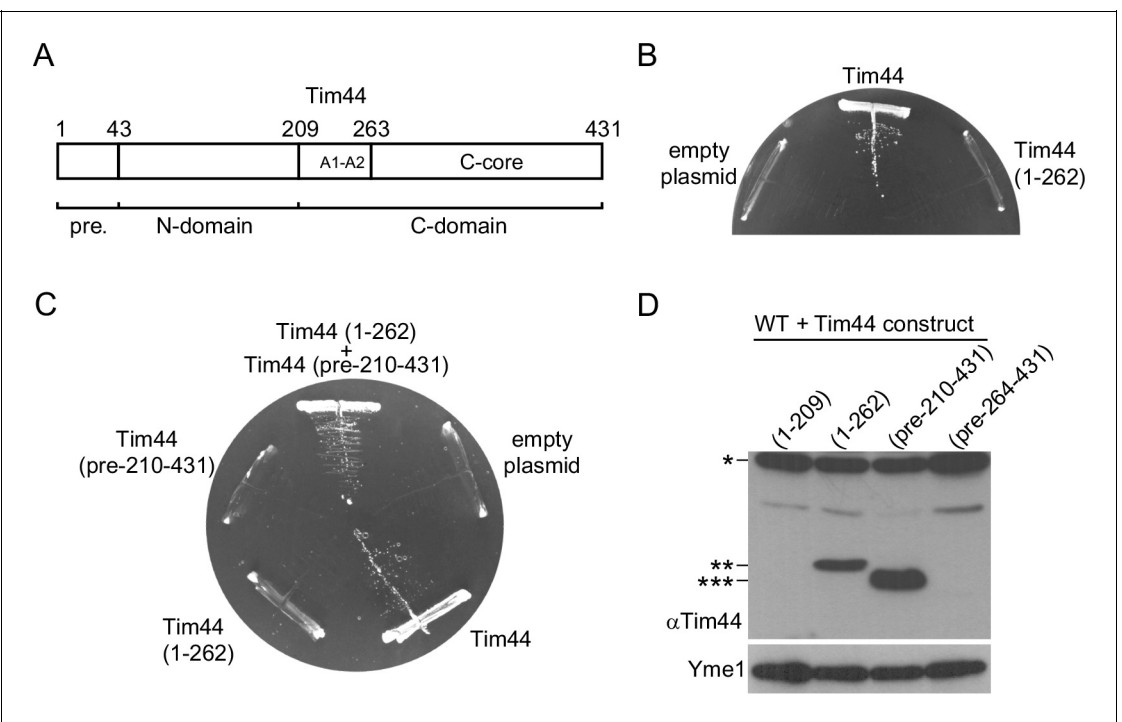

**Figure 1.** The function of Tim44 can be rescued by its two domains expressed *in trans* but not by either of the domains alone. (**A**) Schematic representation of Tim44 domain structure (numbering according to yeast Tim44 sequence). pre. - presequence (**B** and **C**) A haploid yeast deletion strain of *TIM44* carrying the wild-type copy of *TIM44* on a *URA* plasmid was transformed with centromeric plasmids carrying indicated constructs of Tim44 under control of endogenous promoter and 3'UTR. Cells were plated on medium containing 5-fluoroorotic acid and incubated at 30°C. The plasmid carrying wild-type Tim44 and an empty plasmid were used as positive and negative controls, respectively. (**D**) Total cell extracts of wild-type yeast cells transformed with plasmids coding for indicated Tim44 constructs under *GPD* promoter were analysed by SDS–PAGE and immunoblotting against depicted antibodies. *, ** and *** - protein bands detected with antibodies raised against full-length Tim44.

The following figure supplement is available for figure 1:

**Figure supplement 1.** Two domains of Tim44 do not interact stably with each other.

its role in recruitment of Tim44 to cardiolipin-containing membranes (*Weiss et al., 1999*). Based on the crystal structure of the C-terminal domain, a surface-exposed hydrophobic cavity was initially suggested to be important for membrane recruitment (*Josyula et al., 2006*). However, subsequent biochemical studies combined with molecular dynamics simulations, demonstrated that the helices A1 and A2 (residues 235–262 in yeast Tim44), present in the beginning of the C-terminal domain, are important for membrane recruitment (*Marom et al., 2009*). Deletion of helices A1 and A2 abolished membrane association of the C-terminal domain. Interestingly, attachment of helices A1 and A2 to a soluble protein was sufficient to recruit it to a model membrane (*Marom et al., 2009*).

We report here that the function of the full-length Tim44 cannot be rescued by its N-terminal domain extended to include membrane-recruitment helices of the C-terminal domain, demonstrating an unexpected essential function of the core of the C-terminal domain. Surprisingly, we observed that the two domains of Tim44, when expressed *in trans*, can support, although poorly, growth of yeast cells, giving us a tool to dissect the role of the C-terminal domain *in vivo*. We identify the C-terminal domain of Tim44 as the domain of Tim44 that is in contact with translocating proteins and that directly interacts with Tim17, a component of the translocation channel. Our data suggest that intricate rearrangements of the two domains of Tim44 are required during transfer of translocating precursor proteins from the channel in the inner membrane to the ATP-dependent motor at the matrix face.

## Results

### The function of Tim44 can be rescued by its two domains expressed *in trans*

We reasoned that if all important protein–protein interactions of Tim44 are mediated by its N-terminal domain and the only function of the C-terminal domain is to recruit Tim44 to the membrane, then a construct consisting of the N-terminal domain, extended to include the membrane-recruitment helices A1 and A2, should suffice to support the function of the full-length protein. To test this hypothesis, we cloned such a construct in a yeast expression plasmid and transformed it into a Tim44 plasmid shuffle yeast strain. Upon incubation of transformed cells on a medium containing 5-fluoroorotic acid to remove the *URA* plasmid carrying the wild-type, full-length copy of Tim44, no viable cells were obtained (*Figure 1B*). A plasmid carrying the full-length copy of Tim44 enabled growth of yeast cells, whereas no viable colonies were obtained when an empty plasmid was used, confirming the specificity of the assay. We conclude that the N-terminal domain of Tim44, even when extended to include the membrane-recruitment helices of the C-terminal domain, is not sufficient to support the function of the full-length protein. Furthermore, this result suggests that the C-terminal domain of Tim44 has a function beyond membrane recruitment that is apparently essential for viability of yeast cells.

We then tested whether the function of Tim44 can be rescued by its two domains expressed *in trans*. Two plasmids, each encoding one of the two domains of Tim44 and both including A1 and A2 helices, were co-transformed into a Tim44 plasmid shuffle yeast strain and analyzed as above. Surprisingly, we obtained viable colonies when both domains were expressed in the same cell but not when either of the two domains was expressed on its own (*Figure 1C*). The rescue was dependent on the presence of A1 and A2 helices on both domains (data not shown), as in their absence neither of the domains could even be stably expressed in yeast (*Figure 1D*).

It is possible that the two domains of Tim44, both carrying A1 and A2 helices, bind to each other with high affinity and therefore are able to re-establish the full-length protein from the individual domains. To test this possibility, we expressed both domains recombinantly, purified them and analyzed, in a pull down experiment, if they interact with each other. The N-terminally His-tagged N-terminal domain efficiently bound to NiNTA-agarose beads under both low- and high-salt conditions (*Figure 1—figure supplement 1A*). However, we did not observe any copurification of the non-tagged C-terminal domain. We also did not observe any stable interaction of the two domains when digitonin-solubilized mitochondria containing a His-tagged version of the N-terminal domain were used in a NiNTA pull-down experiment (*Figure 1—figure supplement 1B*). Thus, the two domains of Tim44 appear not to stably interact with each other.

## N+C cells are viable, but grow only very poorly even on fermentable medium

We compared growth rate of the yeast strain carrying the wild-type, full-length version of Tim44 (FL) with that of the strain having two Tim44 domains, both containing A1 and A2 helices, expressed *in trans*, for simplicity reasons named from here on N+C. The N+C strain was viable and grew relatively well on a fermentable carbon source at 24°C and 30°C (*Figure 2A*). Still, its growth was slower than that of the FL strain at both temperatures. At 37°C, the N+C strain was barely viable. On a nonfermentable carbon source, when fully functional mitochondria are required, N+C did not grow at any

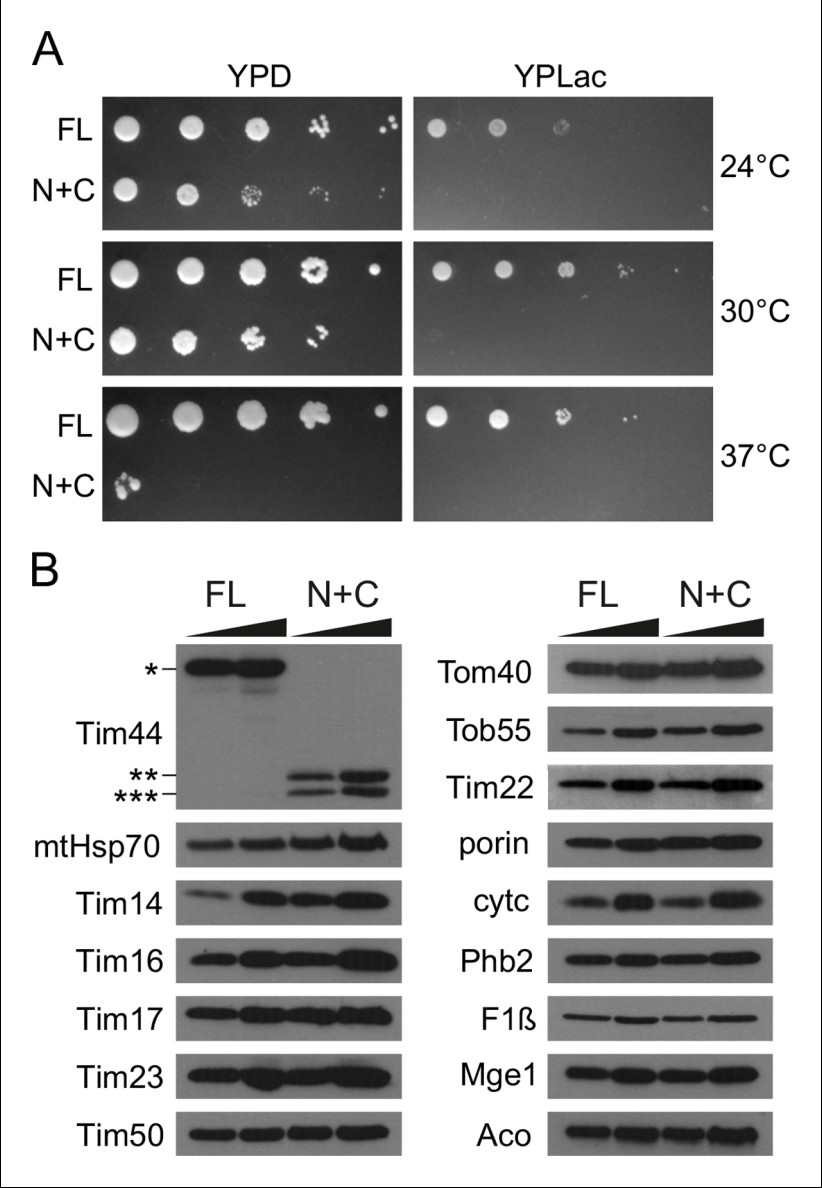

**Figure 2.** N+C cells grow poorly, even on fermentable carbon source. (**A**) Ten-fold serial dilutions of △*tim44* cells rescued by the wild-type, full-length copy of Tim44 (FL) or by its two domains expressed *in trans* (N+C) were spotted on rich medium containing glucose (YPD) or lactate (YPLac), as fermentable and non-fermentable carbon sources, respectively. Plates were incubated at indicated temperatures for 2 (YPD) or 3 days (YPLac). (**B**) 15 and 35 μg of mitochondria isolated from FL and N+C cells were analyzed by SDS–PAGE, followed by immunoblotting against depicted mitochondrial proteins.

of the temperatures tested. Thus, the function of Tim44 can be reconstituted from its two domains separately, although only very poorly.

We isolated mitochondria from FL and N+C strains grown on fermentable medium and compared their mitochondrial protein profiles. Immunostaining with antibodies raised against full-length Tim44 detected no full-length protein in N+C mitochondria but rather two faster migrating bands (*Figure 2B*). Based on the running behavior of the individual domains seen in *Figure 1D*, the slower migrating band corresponds to the N domain and the faster migrating one to the C domain. This confirms that, surprisingly, the full-length Tim44 is indeed not absolutely required for viability of yeast cells. The endogenous levels of other components of the TIM23 complex were either not changed at all (Tim17, Tim23, and Tim50), or were slightly upregulated (mtHsp70, Tim14, and Tim16), likely to compensate for only poorly functional Tim44. Levels of components of other essential mitochondrial protein translocases of the outer and inner mitochondrial membranes, Tom40, Tob55, and Tim22, were not altered compared to FL mitochondria. Similarly, we observed no obvious differences in endogenous levels of proteins present in the outer membrane, intermembrane space, inner membrane, and the matrix that we analyzed.

We conclude that Tim44 can be split into its two domains that are sufficient to support the function of the full-length protein, although only poorly.

## Protein import into mitochondria is severely impaired in N+C cells

Considering the essential role of Tim44 during translocation of precursor proteins into mitochondria, we tested whether the severe growth defect of the N+C strain is due to compromised mitochondrial protein import. When import of precursor proteins into mitochondria is impaired, a precursor form of matrix-localized protein Mdj1 accumulates *in vivo* (*Waegemann et al., 2015*; *Wrobel et al., 2015*). We indeed observed a very prominent band of the precursor form of Mdj1 in total cell extracts of N+C cells, grown at 24°C and 30°C, that was absent in cells containing full-length Tim44 (*Figure 3A*). Thus, the efficiency of protein import into mitochondria is reduced in N+C cells.

To analyze protein import in N+C mitochondria in more detail, we performed *in vitro* protein import into isolated mitochondria (*Figure 3B–G,I–J*). To this end, various mitochondrial precursor proteins were synthesized *in vitro* in the presence of [$^{35}$S]-methionine and incubated with isolated mitochondria. The import efficiencies of all matrix-targeted precursors analyzed, pF1β, pcyt$b_2$(1–167)△DHFR, and pSu9(1–69)DHFR, were drastically reduced in N+C mitochondria when compared to wild type. Import of presequence-containing precursor of Oxa1 that contains multiple transmembrane segments was similarly impaired. Likewise, precursor proteins that are laterally inserted into the inner membrane by the TIM23 complex, such as pDLD1 and pcyt$b_2$, were imported with reduced efficiency into N+C mitochondria. In agreement with the established role of Tim44 in import of precursors of a number of components of respiratory chain complexes and their assembly factors, we observed a slightly reduced membrane potential in N+C mitochondria as compared to wild type (*Figure 3H*). However, precursors of ATP/ADP carrier and of Tim23, whose imports into mitochondria are not dependent on the TIM23 complex, were imported with similar efficiencies in both types of mitochondria, demonstrating that observed effects are not due to general dysfunction of mitochondria. We conclude that splitting of Tim44 into two domains in N+C cells severely impairs transport of proteins by the TIM23 complex, suggesting that full-length Tim44 is required for efficient import of presequence-containing precursor proteins into mitochondria.

## Both domains of Tim44 assemble into the TIM23 complex

Tim44 is thought to play an important role in connecting the translocation channel and the import motor of the TIM23 complex. We thus reasoned that disassembly of the TIM23 complex in N+C mitochondria might be a reason for its reduced functionality. When wild-type mitochondria are solubilized with digitonin, affinity-purified antibodies to Tim17 and to Tim23 essentially deplete both Tim17 and Tim23 from the mitochondrial lysate and precipitate part of Tim50, Tim44, Tim14, and Tim16 (*Figure 4*). Similarly, affinity-purified antibodies to Tim16 deplete both Tim16 and Tim14 and precipitate Tim50, Tim17, Tim23, and Tim44 from mitochondrial lysate. We observed essentially the same precipitation pattern when we analyzed digitonin-solubilized N+C mitochondria, demonstrating that the TIM23 complex is properly assembled. Importantly, both N and C domains of Tim44 were recruited to the TIM23 complex.

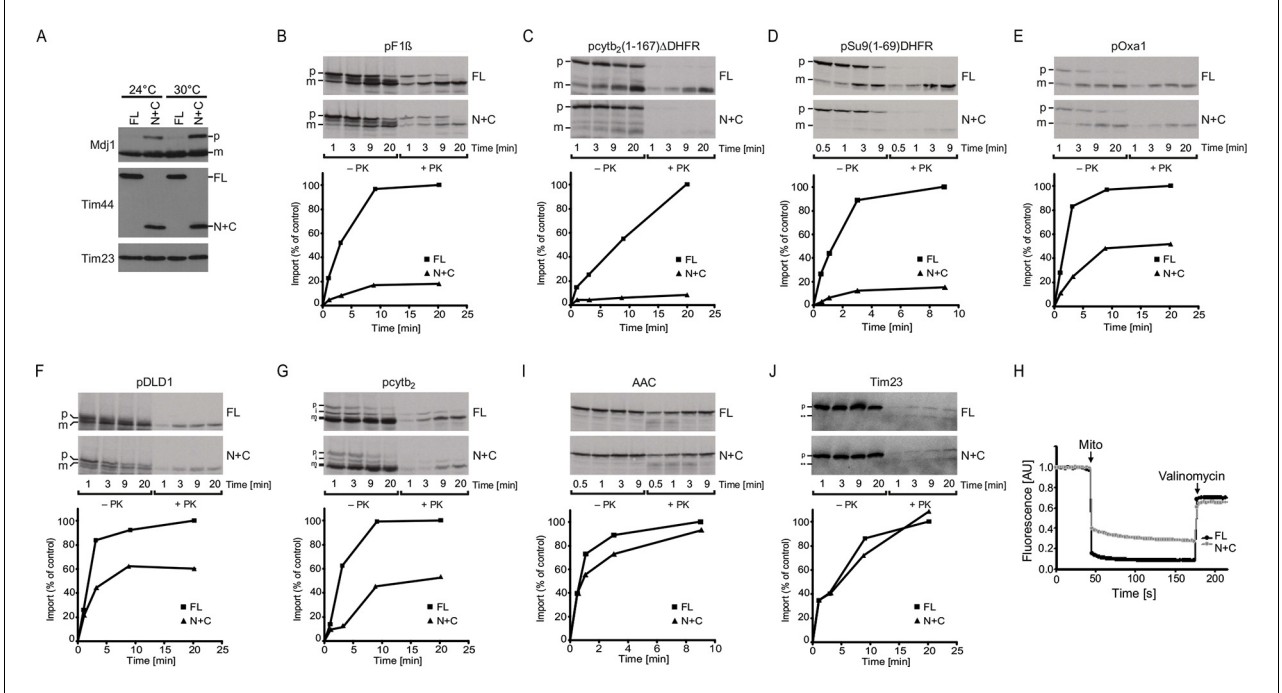

**Figure 3.** N+C cells have a strongly impaired import via the TIM23 complex. (**A**) Total cell extracts of FL and N+C cells grown at 24°C and 30°C were analyzed by SDS–PAGE and immunoblotting using indicated antibodies. p - precursor, and m - mature form of Mdj1. (**B–G** and **I–J**) $^{35}$S-labeled mitochondrial precursor proteins were imported into mitochondria isolated from FL and N+C cells. After indicated time periods, aliquots were removed and Proteinase K (PK) was added where indicated. Samples were analyzed by SDS–PAGE, autoradiography and quantification of PK-protected mature forms of imported proteins. pF1β - precursor of the β subunit of $F_oF_1$ ATPase. pcyt$b_2$(1–167)△DHFR - precursor consisting of the first 167 residues with the deleted sorting signal of yeast cytochrome $b_2$ fused to mouse dihydrofolate reductase (DHFR); pSu9(1–69)DHFR - matrix targeting signal (residues 1–69) of subunit 9 of $F_oF_1$ ATPase from *Neurospora crassa* fused to DHFR; pOxa1 - precursor of Oxa1; pDLD1 - precursor of D-lactate dehydrogenase; pcyt$b_2$ - precursor of cytochrome $b_2$; AAC - precursor of ATP/ADP carrier; p, i, m - precursor, intermediate, and mature forms of imported proteins; * - *in vitro* translation product starting from an internal methionine. ** - clipped form of Tim23. (**H**) Membrane potential of isolated mitochondria was measured using DiSC$_3$(5). Valinomycin was added to dissipate membrane potential.

## The TIM23 complex adopts an altered conformation in N+C mitochondria

Since the assembly of the TIM23 complex is not affected in N+C mitochondria, we reasoned that an altered conformational flexibility may be a reason behind its reduced function in N+C cells. Chemical crosslinking is currently the most sensitive assay available to analyze the conformation of the TIM23 complex in intact mitochondria. We thus compared the crosslinking patterns of TIM23 subunits in N+C mitochondria to those in FL. In wild-type mitochondria, Tim16 can be crosslinked to mtHsp70, Tim44, and Tim14 in an ATP-dependent manner (*Figure 5A*). In N+C mitochondria, the same crosslinks of Tim16 to mtHsp70 and to Tim14 were observed. The crosslink to Tim44 was, as expected, absent in N+C mitochondria and another crosslink to a smaller protein appeared. In addition, a crosslink between two Tim16 molecules became prominent. Interestingly, this crosslink has previously been observed in mutants in which conformation of the TIM23 complex was altered (*Popov-Čeleketić et al., 2008*). Similarly, we observed prominent changes in crosslinking pattern of the channel component Tim23 (*Figure 5B*). In addition to the crosslink of Tim23 to Pam17, observed in both FL and N+C mitochondria, a prominent Tim23-dimer crosslink appeared in N+C mitochondria.

To obtain an independent evidence that the conformation of the TIM23 complex is affected in N+C mitochondria, we analyzed the complex by blue native gel electrophoresis. When digitonin-solubilized wild-type mitochondria are separated by BN-PAGE, Tim17, and Tim23 are present in a 90 kDa complex and, to a lesser degree, in higher molecular weight complexes that additionally contain Tim21 and Mgr2 (*Chacinska et al., 2005*; *Ieva et al., 2014*). In contrast, with digitonin-solubilized N+C mitochondria, antibodies to Tim17 and Tim23 revealed slightly shifted bands, in particular

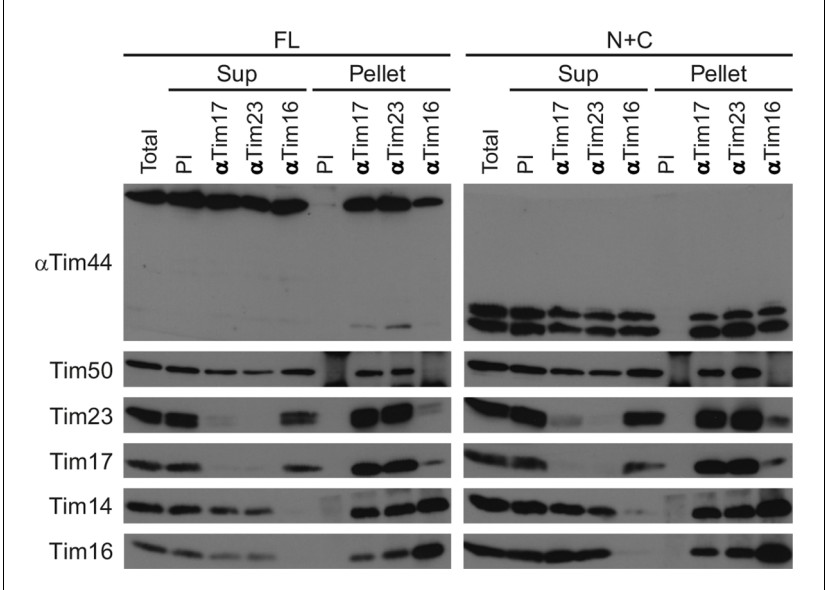

**Figure 4.** The TIM23 complex is assembled in N+C mitochondria. Mitochondria from FL and N+C cells were solubilized with digitonin-containing buffer and mitochondrial lysates incubated with affinity-purified antibodies to Tim17, Tim23, and Tim16 prebound to Protein A-Sepharose beads. Antibodies from preimmune serum (PI) were used as a negative control. After three washing steps, material specifically bound to the beads was eluted with Laemmli buffer. Total (20%), supernatant (Sup, 20%), and bound (Pellet, 100%) fractions were analyzed by SDS–PAGE and immunoblotting with indicated antibodies.

of the 90 kDa complex (*Figure 5C*). Since the 90 kDa complex does not contain any other known subunit of the TIM23 complex, this finding further supports the above notion that the conformation of the translocation channel is changed in N+C mitochondria. We observed no obvious difference in the ca. 60 kDa Tim14-Tim16 complex between FL and N+C mitochondria. As expected, full-length Tim44, present in FL mitochondria, was absent in N+C mitochondria (*Figure 5C*).

Together, these results demonstrate that the conformation of the TIM23 complex is changed in N+C mitochondria. They further show that alterations in the components traditionally assigned to the import motor affect the conformation of the translocation channel in the inner membrane, supporting the notion of an intricate crosstalk within the complex.

## Role of the C-terminal domain of Tim44

The data presented so far suggest that full-length Tim44 is required for optimal conformational dynamics of the TIM23 complex. Furthermore, they suggest that the C-terminal domain has an essential function within the TIM23 complex, beyond mere membrane recruitment. So, what is the function of the C-terminal domain of Tim44? We first searched for binding partners of the individual domains. To that end, we recombinantly expressed and purified full-length Tim44 as well as its two domains (*Figure 6A*). To look for interaction partners of the core domains, both domains now lacked the segment containing A1 and A2 helices. Purified proteins were covalently coupled to the Sepharose beads and were subsequently incubated with mitochondrial lysates. Mitochondria were solubilized with Triton X-100 that, unlike digitonin, dissociates the TIM23 complex into its individual subunits (except for the Tim14-Tim16 subcomplex that remains stable). In this way, direct protein-protein interactions can be analyzed. We observed prominent, specific binding of mtHsp70, Tim16, Tim14 and Tim17, and to a far lesser degree of Tim23 and Tim50, to full-length Tim44 (*Figure 6B*). None of the proteins bound to empty beads. Also, we observed no binding of two abundant mitochondrial proteins, porin, and F1βß, demonstrating the specificity of observed interactions. mtHsp70, Tim16 and Tim14 also efficiently bound to the N-terminal domain of Tim44, in agreement with previous observations (*Schilke et al., 2012*; *Schiller et al., 2008*), and far less efficiently to the C-terminal domain. Since the Tim14-Tim16 subcomplex remains stable in Triton X-100, it is not

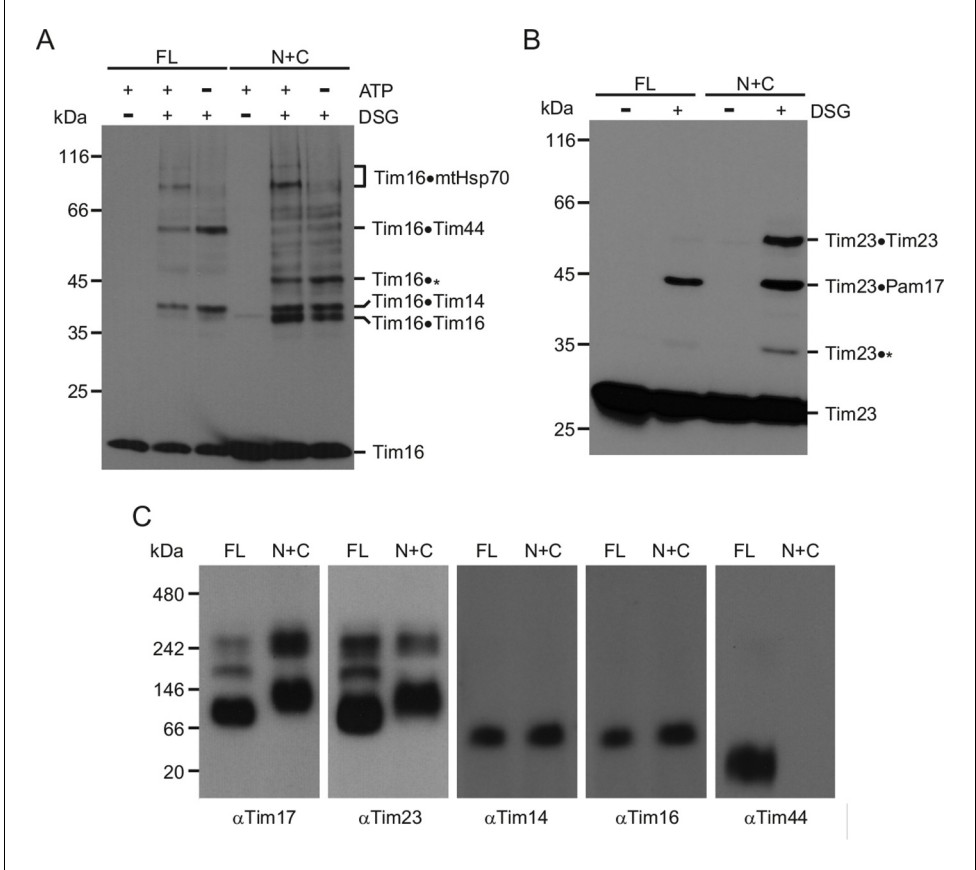

**Figure 5.** The TIM23 complex adopts an altered conformation in N+C mitochondria. (**A** and **B**) Mitochondria from FL and N+C cells were incubated with amino group-specific crosslinker disuccinimidyl glutarate (DSG). Where indicated, mitochondrial ATP levels were altered prior to crosslinking. After quenching of excess crosslinker, mitochondria were reisolated and analyzed by SDS–PAGE followed by immunoblotting with antibodies to Tim16 (**A**) and Tim23 (**B**). * indicates currently uncharacterized crosslinks. (**C**) Mitochondria from FL and N+C cells were solubilized in digitonin-containing buffer and analyzed by BN-PAGE and immunoblotting with indicated antibodies.

possible by this method to distinguish which of the two subunits, or maybe even both, directly interacts with the N-terminal domain of Tim44. Binding of Tim17 to the N-terminal domain of Tim44 was drastically lower compared to its binding to the full-length protein. Instead, a strong binding of Tim17 to the C-terminal domain of Tim44 was observed.

We conclude that the N-terminal domain of Tim44 binds to the components of the import motor, whereas the C-terminal domain binds to the translocation channel in the inner membrane, revealing a novel function of the C-terminal domain of Tim44.

We then asked which of the two domains of Tim44 is in contact with translocating proteins. To answer this question, we first affinity-purified antibodies that specifically recognize cores of the individual domains of Tim44 using the above described Sepharose beads. The antibodies, affinity purified using beads with coupled full-length Tim44, recognized full-length Tim44 as well as both of its domains (*Figure 6C*). In contrast, antibodies that were affinity purified using beads with coupled individual domains recognized only the respective domain and the full-length protein (*Figure 6C*). This demonstrates that we indeed purified antibodies specific for individual domains of Tim44. Next, we accumulated [35]S-labelled precursor protein pcyt$b_2$(1–167)△DHFR as a TOM-TIM23-spanning intermediate. Briefly, this precursor protein consists of the first 167 residues of yeast cytochrome $b_2$, with a 19 residue deletion in its lateral insertion signal, fused to the passenger protein dihydrofolate reductase. In the presence of methotrexate, that stabilizes folded DHFR, the $b_2$ part reaches the matrix, whereas the DHFR moiety remains on the mitochondrial surface resulting in an intermediate that spans both TOM and TIM23 complexes. The association of Tim44 and its domains with the

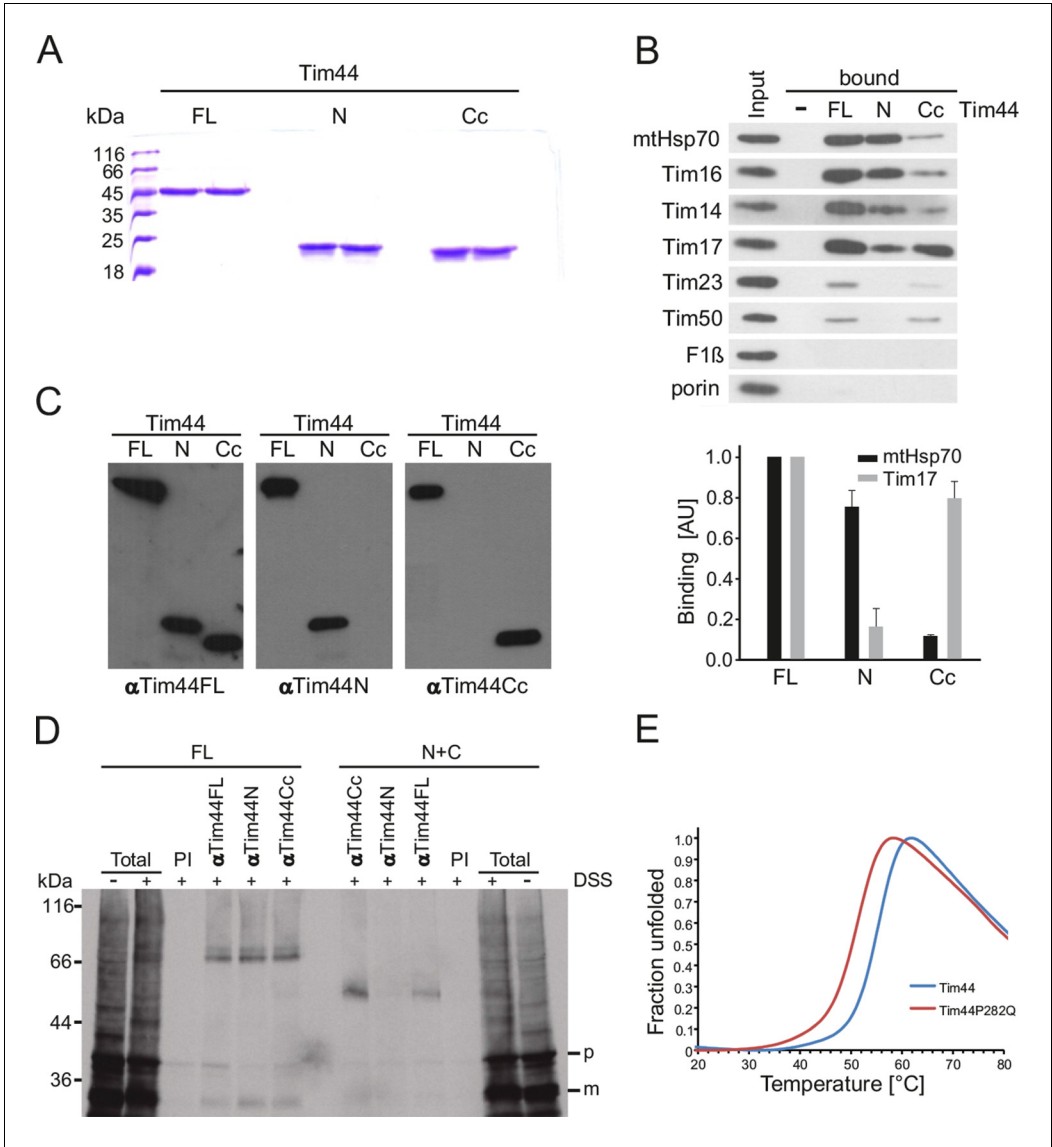

**Figure 6.** C-terminal domain of Tim44 interacts with Tim17 and with a precursor in transit. (**A**) Coomassie-stained SDS-PA gel of recombinantly expressed and purified constructs of Tim44. FL - full-length, mature Tim44 (residues 43–431); N - a construct encompassing the N-terminal domain of Tim44 (residues 43–209); Cc - a construct encompassing the core of the C-terminal domain of Tim44 (residues 264–431). (**B**) Wild-type mitochondria were solubilized with Triton X-100 and incubated with indicated purified constructs of Tim44 covalently coupled to CNBr-Sepharose beads. Beads with no coupled protein were used as a negative control. After washing steps, proteins specifically bound to the beads were eluted by Laemmli buffer and analyzed by SDS–PAGE followed by immunoblotting with the indicated antibodies. Input lane contains 4.5% of the material used for binding (upper panel). Binding of mtHsp70, as a representative of the import motor components, and of Tim17 to different beads was quantified from three independent experiments (lower panel). Binding to FL was set to 1. (**C**) Antibodies specific for N and Cc domains of Tim44 were affinity purified from rabbit serum raised against full-length Tim44 using respective domains of Tim44 covalently coupled to Sepharose beads, as described under (**B**). To test the specificity of purified antibodies, indicated Tim44 constructs were loaded on an SDS-PA gel, blotted on a nitrocellulose membrane and obtained membranes were immunoblotted using the purified antibodies, as indicated. (**D**) $^{35}$S-labelled matrix targeted precursor protein pcyt$b_2$(1–167)ΔDHFR was imported into isolated mitochondria from FL and N+C cells in the presence of methotrexate, leading to its arrest as a TOM-TIM23 spanning intermediate. Samples were then crosslinked with disuccinimidyl suberate (DSS), where indicated. After quenching of excess crosslinker, aliquots were taken out for 'total' and the rest of samples solubilized in SDS-containing buffer to dissociate all noncovalent protein–protein interactions. Solubilized material was incubated with indicated affinity-purified antibodies prebound to Protein A-Sepharose beads. Antibodies from preimmune serum (PI) were used as a negative control. Material specifically bound to the beads was eluted with Laemmli buffer and analyzed by SDS–PAGE and autoradiography. p - precursor and m - mature forms of pcyt$b_2$(1–167)ΔDHFR. (**E**) Melting curves of recombinant wild type and Pro282Gln mutant of Tim44 obtained by thermal shift assay.

arrested precursor protein was analyzed by chemical crosslinking followed by immunoprecipitation with antibodies to full-length Tim44 and its individual domains. In wild-type mitochondria, all three antibodies precipitated a crosslinking adduct of Tim44 to the arrested precursor protein, demonstrating that they are all able to immunoprecipitate the respective antigens (*Figure 6D*). In contrast, with N+C mitochondria, a faster migrating crosslinking adduct of a Tim44 domain to the arrested precursor protein was immunoprecipitated with the antibodies against the C-terminal domain and against the full-length protein but not with the antibodies against the N-terminal domain. This demonstrates that the C-terminal domain of Tim44 is in close vicinity of the translocating protein.

Mutations identified in human patients can frequently point to functionally important residues in affected proteins. In this respect, Pro308Gln mutation in human Tim44 has recently been linked to oncocytic thyroid carcinoma (*Bonora et al., 2006*). Since the mutation maps to the C-terminal domain of Tim44, we wanted to analyze functional implications of this mutation and therefore made the corresponding mutation in yeast Tim44 (Pro282Gln). We compared thermal stabilities of wild type and mutant Tim44 proteins by thermal shift assay. The melting temperature of wild-type Tim44 was 54°C, whereas that of the mutant protein was 4°C lower (*Figure 6E*). This demonstrates that the mutation significantly destabilizes Tim44, providing first clues toward molecular understanding of the associated human disease.

## Discussion

The major question of protein import into mitochondria that has remained unresolved is how translocation of precursor proteins through the channel in the inner membrane is coupled to the ATP-dependent activity of the Hsp70-based import motor at the matrix face of the inner membrane.

Results presented here demonstrate that the two domain structure of Tim44 is essential during this process. We show here that the two domains of Tim44 have different interaction partners within the TIM23 complex. In this way, Tim44 holds the TIM23 complex together. Our data revealed a direct, previously unexpected interaction between the C-terminal domain of Tim44 with the channel component Tim17. This result not only assigned a novel function to the C-terminal domain of Tim44 but also shed new light on Tim17, the component of the TIM23 complex that has been notoriously difficult to analyze. Recent mutational analysis of the matrix exposed loop between transmembrane segments 1 and 2 of Tim17 revealed no interaction site for Tim44 (*Ting et al., 2014*), suggesting its presence in another segment of the protein. Our data also confirmed the previously observed interactions of the N-terminal domain of Tim44 with the components of the import motor (*Schilke et al., 2012*; *Schiller et al., 2008*). We did, however, not observe any direct interaction between Tim23 and the N-terminal domain of Tim44 that has previously been seen by crosslinking in intact mitochondria (*Ting et al., 2014*). It is possible that this crosslinking requires a specific conformation of Tim23 only adopted when Tim23 is bound to Tim17 in the inner membrane. This notion is supported by our previous observation that the stable binding of Tim44 to the translocation channel requires assembled Tim17-Tim23 core of the TIM23 complex (*Mokranjac et al., 2003b*). We observed a direct Tim17-Tim44 interaction here probably because of a high local concentration of the C-terminal domain when bound to the beads.

The core of the C-terminal domain is preceded by a segment that contains two amphipathic, membrane-recruitment helices. This central segment connects the two domains of Tim44. Intriguingly, the two currently available crystal structures of the C-terminal domains of yeast and human Tim44s showed different orientations of the two helices relative to the core domains (*Handa et al., 2007*; *Josyula et al., 2006*). The conformational change was likely induced upon PEG binding to this region of human Tim44 during crystallization (*Handa et al., 2007*). It is tempting to speculate that the same conformational change takes place during translocation of proteins in the mitochondria. Such a conformational change would not only reorient the two helices in respect to the core of the C-domain but also change the relative orientation of N- and C-terminal domains. Since the two domains have different interaction partners within the TIM23 complex, such a change could rearrange the entire complex. The importance of this proposed conformational change in Tim44 is supported by the data presented here. The function of the full-length Tim44 could be reconstituted from its individual domains only very poorly. Also, there is obviously a very strong evolutionary pressure to keep the two domains of Tim44 within one polypeptide chain. N+C strain had to be kept at all times on the selective medium - even after only an overnight incubation on a nonselective

medium the full-length protein reappeared (our unpublished observation), likely due to a recombination event between two plasmids.

Tim44 can be crosslinked to translocating proteins. Our data revealed that it is the C-terminal domain of Tim44 that interacts with proteins entering the matrix from the translocation channel in the inner membrane. A direct interaction of the same domain with Tim17 would optimally position the C-terminal domain to the outlet of the translocation channel. This raises an interesting possibility that translocating precursor proteins may play an important role in the above postulated conformational changes of Tim44.

A missense mutation Pro308Gln in human Tim44 is associated with familial oncocytic thyroid carcinoma. The corresponding mutation in yeast, Pro282Gln, destabilized the protein but produced no obvious growth phenotype or an *in vivo* import defect (our unpublished observations), suggesting that the yeast system is more robust. This observation is in agreement with the notion that mutations that would severely affect the function of the TIM23 complex would likely be embryonically lethal in humans. Still, the disease caused by a mutation in the C-terminal domain of human Tim44 speaks for an important role of this domain in the function of the entire TIM23 complex. Furthermore, the mutation maps to the short loop between A3 and A4 helices in the C-terminal domain of Tim44. Based on the crystal structure of Tim44, it was previously suggested that the mutation could affect the conformational flexibility of the A1 and A2 helices (*Handa et al., 2007*), intriguingly providing further support for the above postulated conformational changes of Tim44.

Based on the previously available data and the results presented here, we put forward the following model to describe how translocation of precursor proteins through the channel in the inner membrane is coupled to their capture by the ATP-dependent import motor at the matrix face of the channel (*Figure 7*). Tim44 plays a central role in this model. We envisage that two domains of Tim44

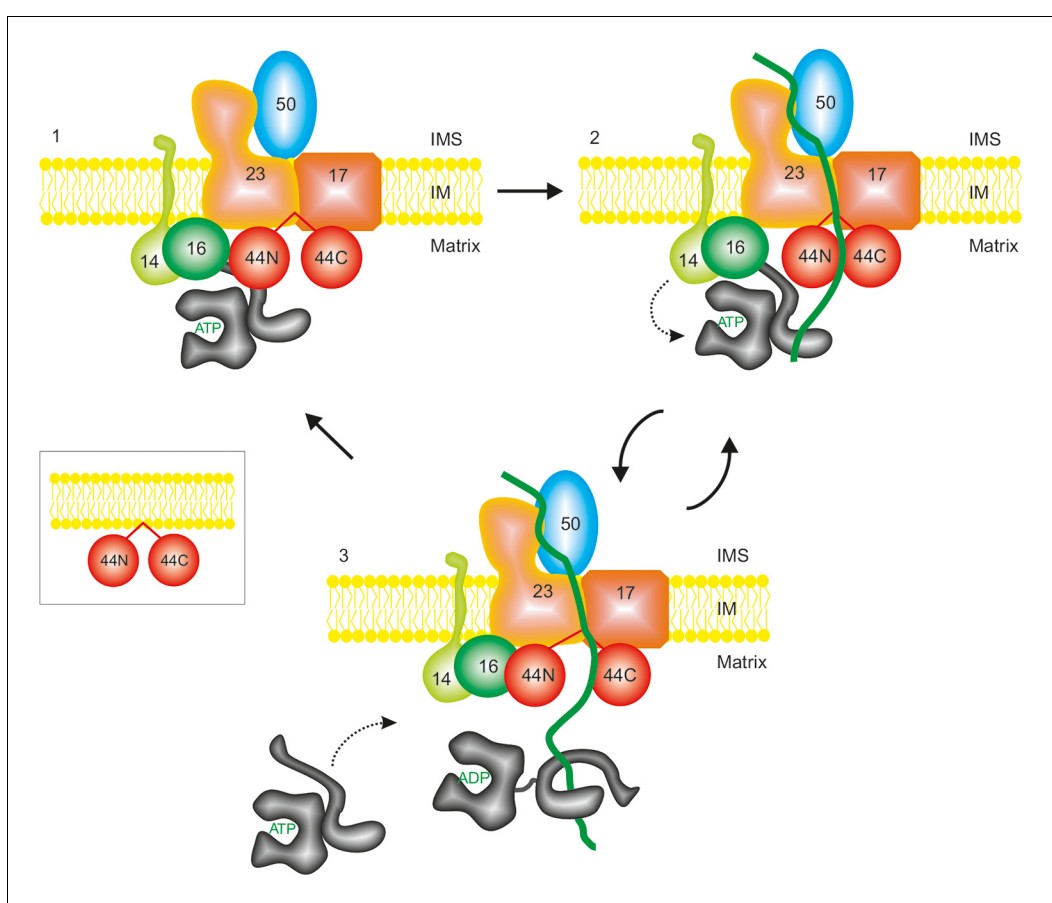

**Figure 7.** A proposed model of function of the TIM23 complex. See text for details. For simplicity reasons, only essential subunits of the complex are shown.

are connected by the central segment that contains membrane-recruitment helices, like two cherries on the stalks (*Figure 7* insert). This central segment of Tim44 recruits the protein to the cardiolipin-containing membranes. There, through direct protein–protein interactions, the C-terminal domain of Tim44 binds to Tim17 and the N-terminal domain to mtHsp70 and to Tim14-Tim16 subcomplex (1). In this way, Tim44 functions as a central platform that connects the translocation channel in the inner membrane with the import motor at the matrix face. Additional interactions likely stabilize the complex, in particular that between the N-terminal domain of Tim44 and Tim23 (*Ting et al., 2014*) as well as the one between Tim17 and the IMS-exposed segment of Tim14 (*Chacinska et al., 2005*). In the resting state, the translocation channel is closed to maintain the permeability barrier of the inner membrane. During translocation of proteins (2), the translocation channel in the inner membrane has to open to allow passage of proteins. Opening of the channel will likely change the conformation of Tim17 that could be further conveyed to the C-terminal domain Tim44. It is tempting to speculate that this conformational change is transduced to the N-terminal domain of Tim44 through the central, membrane-bound segment of Tim44, leading to relative rearrangements of the two domains of Tim44. This change would now allow Tim14-Tim16 complex to stimulate the ATPase activity of mtHsp70 leading to stable binding of the translocating protein to mtHsp70. mtHsp70, with bound polypeptide, will then move into the matrix, opening a binding site on Tim44 for another molecule of mtHsp70 (3). We speculate that the release of mtHsp70 with bound polypeptide from the N-terminal domain of Tim44 will send a signal back to the C-terminal domain of Tim44 and further to the translocation channel. Multiple cycles of mtHsp70 are required to translocate the entire polypeptide chain into the matrix. Once the entire polypeptide has been translocated, the translocation channel will revert to its resting, closed state, bringing also Tim44 back to its resting conformation (1). Thus, the translocation channel in the inner membrane and the mtHsp70 system at the matrix face communicate with each other through rearrangements of the two domains of Tim44 that are stimulated by translocating polypeptide chain.

## Material and methods

### Yeast strains, plasmids, and growth conditions

Wild-type haploid yeast strain YPH499 was used for all genetic manipulations. A Tim44 plasmid shuffling yeast strain was made by transforming YPH499 cells with a pVT-102U plasmid (*URA* marker) containing a full-length *TIM44* followed by replacement of the chromosomal copy of *TIM44* with a *HIS3* cassette by homologous recombination. For complementation analyzes, endogenous promoter, mitochondrial presequence (residues 1–42) and the 3′-untranslated region of *TIM44* were cloned into centromeric yeast plasmids pRS315 (*LEU* marker) and pRS314 (*TRP* marker) and obtained plasmids subsequently used for cloning of various Tim44 constructs. The following constructs were used in the analyzes: Tim44(43–209), Tim44(43–262), Tim44(264–431), and Tim44(210–431). The constructs encompassing the N- and the C-terminal domains of Tim44 were cloned into pRS315 and pRS314 plasmids, respectively. Plasmids carrying the full-length copy of *TIM44* were used as positive controls and empty plasmids as negative ones. A Tim44 plasmid shuffling yeast strain was transformed with two plasmids simultaneously and selected on selective glucose medium lacking respective markers. Cells that lost the wild-type copy of Tim44 on the *URA* plasmid were selected on medium containing 5-fluoroorotic acid at 30°C.

For expression in the wild-type background, the above-described constructs of Tim44, containing endogenous Tim44 presequence, were also cloned into centromeric yeast plasmids p414GPD and p415GPD for expression under the control of the strong *GPD* promoter. Cells were grown on selective lactate medium containing 0.1% glucose.

FL and N+C cells were grown in selective glucose medium at 30°C, unless otherwise indicated, and mitochondria were isolated from cells in logarithmic growth phase.

### Recombinant proteins

DNA sequences coding for various segments of Tim44 were cloned into bacterial expression vector pET-Duet1 introducing a TEV cleavage site between the His$_6$-tag and the protein coding region. The following Tim44 constructs were cloned: Tim44(43–431) (full-length protein lacking the mitochondrial presequence), Tim44(43–209) (referred to as N in *Figure 6A*), Tim44(43–263), Tim44(211–431), and

Tim44(264–431) (referred to as Cc in *Figure 6A*). Pro282Gln mutation was introduced into the full-length construct using site directed mutagenesis. Proteins were expressed in *E. coli* BL21(DE3) at 37°C and purified using affinity chromatography on NiNTA-agarose beads (Qiagen, Germany) followed by gel filtration on Superdex 75 column (GE Healthcare, Germany). Unless otherwise indicated, the His$_6$-tags were removed by incubation with the TEV protease. The purified proteins were stored at -80°C in 20 mM HEPES/KOH, 200 mM KCl, 5 mM MgCl$_2$, pH 7.5, until use.

Purified proteins were coupled to CNBr-Sepharose beads (GE Healthcare, Germany) according to manufacturer's instructions and stored at 4°C. The beads were used for purification of domain-specific antibodies from the serum raised in rabbits against recombinantly expressed full-length Tim44. For direct binding analysis, mitochondria isolated from wild-type yeast cells were solubilized with 0.5% Triton X-100 in 20 mM Tris/HCl, pH 8.0, 80 mM KCl, 10% glycerol at 1 mg/mL and incubated with Tim44 constructs coupled to CNBr-Sepharose beads for 30 min at 4°C. After three washing steps, specifically bound proteins were eluted with Laemmli buffer. Samples were analyzed by SDS–PAGE and immunoblotting.

## Thermal shift assay

Thermal stabilities of wild type and P282Q mutant form of Tim44 were analyzed by fluorescence thermal shift assay (*Müller et al., 2015*). Recombinant proteins (6.2 µM) in 20 mM HEPES/NaOH, 150 mM NaCl, pH 7.1 were mixed with 5x SYPRO Orange and melting curves analyzed in a real-time PCR machine using a gradient from 5°C to 99°C. Three technical replicates of two independent protein purifications were analyzed in parallel. Mutant Tim44 showed significantly decreased thermal stability under all conditions analyzed - in buffers containing different salt concentrations (50, 150, and 450 mM) as well as in different buffers and pHs (HEPES buffer at pH 7.1 and phosphate buffer at pH 8.0).

## Miscellaneous

Previously published procedures were used for protein import into isolated mitochondria, crosslinking, coimmunoprecipitations and arrest of mitochondrial precursor proteins as TOM-TIM23 spanning intermediates followed by crosslinking and immunoprecipitation under denaturing conditions (*Mokranjac et al., 2003a*; *2003b*; *Popov-Čeleketić et al., 2008*).

## Acknowledgements

We are grateful to Drs W Neupert, A Bracher, and M Sichting for stimulating discussions. We thank P Robisch and Z Stanic for expert technical assistance, Dr A Azem for careful reading of the manuscript and Dr A Ladurner for his continuous support. This work was supported by Deutsche Forschungsgemeinschaft (MO1944/1-1 to DM and GRK1721 to GW), DAAD PhD fellowship (to RB) and DAAD RISE fellowship (to CG).

## Additional information

### Funding

| Funder | Grant reference number | Author |
| --- | --- | --- |
| Deutscher Akademischer Austauschdienst | Graduate student fellowship | Rupa Banerjee |
| Deutscher Akademischer Austauschdienst | RISE visiting student fellowship | Christina Gladkova |
| Deutsche Forschungsgemeinschaft | GRK1721 | Gregor Witte |
| Deutsche Forschungsgemeinschaft | MO1944/1-1 | Dejana Mokranjac |

The funders had no role in study design, data collection and interpretation, or the decision to submit the work for publication.

## Author contributions

RB, DM, Conception and design, Acquisition of data, Analysis and interpretation of data, Drafting or revising the article; CG, GW, Acquisition of data, Analysis and interpretation of data, Drafting or revising the article; KM, Acquisition of data, Contributed unpublished essential data or reagents

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
