## [Decision Letter]

Thank you for submitting your work entitled "Rearrangements of the two domains of Tim44 drive protein translocation into mitochondria" for consideration by *eLife*. Your article has been favorably evaluated by Randy Schekman (Senior editor) and three reviewers, one of whom, Klaus Pfanner, is a member of our Board of Reviewing Editors. The reviewers have discussed the reviews with one another and the Reviewing editor has drafted this decision to help you prepare a revised submission.

Summary:

Mitochondrial import of matrix proteins requires the mitochondrial import motor to drive protein translocation. The manuscript by Banerjee et al. investigates the structure function relation of Tim44, an essential subunit of the mitochondrial import motor. To dissect the N- and C-terminal domains functionally, the authors expressed these domains in trans and find that they suffice to confer viability to yeast cells lacking TIM44. Their analyses reveal that mitochondria with "split" Tim44 display transport defects along the presequence import route. Their analyses suggest that the N- and C-terminal domains engage with different modules of the presequence translocase. In comparison to full length Tim44, the split protein causes reorganization within the presequence translocase, implicating Tim44 in translocase/motor communication.

This is an interesting analysis that provides new and unexpected insight into how the import motor constituent Tim44 engages with the presequence translocase. Moreover, the data provides evidence for a link between motor dynamics and translocase organization. Thus, the manuscript addresses a long standing question on how the dynamic of the import motor is linked to translocase function.

Essential revisions:

1) Since rearrangements of the two domains of Tim44 are not directly demonstrated, the title should be modified to better reflect the main message of the paper.

2) The observed import defects of sorted and matrix targeted precursors is surprising. Why are sorted proteins affected that should not depend on the ATP driven motor and thus not on Tim44 function? The authors should assess the membrane potential directly to exclude indirect effects. Since the membrane potential-dependence of precursors varies a lot, AAC import alone might be misleading. In addition, carrier transport should be analyzed also by BN PAGE.

3) Ting et al. (2014) reported that the N-terminal domain of Tim44 interacts with Tim23 as well as motor components by using crosslinking in intact mitochondria. Using purified Tim44 domains and mitochondria lysed with detergent, the authors confirmed the interaction of the N-terminal domain with motor components, however, not with Tim23. They speculate that crosslinking of the N-terminal Tim44 domain to Tim23 requires a specific conformation of Tim23 only adopted when Tim23 is bound to Tim17 in the inner membrane (i.e. in intact mitochondria). Since the authors demonstrate the interaction of the C-terminal domain of Tim44 with Tim17 by using the purified domains and lysed mitochondria, but use crosslinking in intact mitochondria for showing the interaction with precursor proteins, the two approaches should be compared more directly. A suitable approach would be to analyze the interaction of the C-terminal domain of Tim44 with Tim17 in intact mitochondria by crosslinking. They may use radiolabeled Tim17 imported into mitochondria and their domain-specific Tim44 antibodies to address this topic.

4) The purification experiments shown in Figure 6 and Figure 4 will benefit from quantification/statistics. In case of Figure 6, the outcome is clearly visible in the presented experiment, but this seems to be only a representative (of how many experiments?). The statistical analysis of repetitions would provide more confidence for this important conclusion. In the case of Figure 4, the conclusion about the presence of the N-and C-domains is clear looking at the presented experiment (representative?), in contrast to the conclusion concerning Tim21 and Pam17. If the authors would like to keep the observation concerning Tim21 and Pam17 in the text, it requires statistics.

Figure 4: It is difficult to compare the co-immunoprecipitation efficiencies of different TIM23 components since the membranes have been strongly trimmed and the blots are overexposed. Here different exposures might have to be presented.

5) Figure 4: There is far less Tim44 N+C associated with the TIM23 complex than full-length Tim44 in WT. This raises the question to what extend a Tim44-depletion mirrors the observed effects for the split protein with regard to crosslinks that do not include Tim44 (Tim23-Tim23, Tim16-Tim16) or import defects of sorted proteins.

6) The conclusion that the TIM23 complex adopts an "altered conformation" in N+C mitochondria is based on crosslinking experiments. It would be good to support this conclusion by an independent approach, such as native gel electrophoresis or differential protease sensitivity of the TIM23 components.

Moreover, the term "altered conformational flexibility" is not fully fitting to the results presented here. The authors rather look at an altered subunit composition and not protein conformations. The text should be changed.

7) The part on the human patient mutation mimicked in yeast Tim44 is interesting, but not required for the rest of the story. Does the mutation in Tim44 affect the TIM23 complex or protein import in yeast or humans? As it stands it is not very informative. The authors should either remove that part or perform a functional analysis, i.e. test mutant cells/mitochondria for the functionality of the Tim44 C-terminal domain in the binding of Tim17 and precursor binding during mitochondrial import.

---

## [Author Response]

*1) Since rearrangements of the two domains of Tim44 are not directly demonstrated, the title should be modified to better reflect the main message of the paper.*

The title was modified to now read “Two domains of Tim44 have different roles in the presequence translocase during transport of proteins into mitochondria”. We agree that the title may have been somewhat misleading as we have indeed not directly demonstrated rearrangements of the two domains of Tim44.

*2) The observed import defects of sorted and matrix targeted precursors is surprising. Why are sorted proteins affected that should not depend on the ATP driven motor and thus not on Tim44 function? The authors should assess the membrane potential directly to exclude indirect effects. Since the membrane potential-dependence of precursors varies a lot, AAC import alone might be misleading. In addition, carrier transport should be analyzed also by BN PAGE.*

We would not agree that the import defects of TIM23 substrates we observed in N+C mitochondria are in general surprising. Tim44 has been identified over twenty years ago as an essential component of the TIM23 complex required for transport of presequence-containing precursors into mitochondria. Over the years, several temperature-sensitive mutants of Tim44 have been identified that specifically affected transport along the TIM23 pathway. We would thus rather argue that it would be surprising if in N+C mitochondria, which contain only a split version of this protein, no import defect was observed. We were certainly surprised to see that the N+C cells are viable at all. The point that we observed no difference between sorted and matrix targeted precursors may potentially be seen as surprising. However, it should also be said that the role of Tim44 in the differential sorting of proteins by the TIM23 complex has not been carefully analyzed, at least to the best of our knowledge, in contrast to mtHsp70 and the J proteins Tim14/Pam18 and Tim16/Pam16. While the latter proteins are very likely directly involved in the ATP-dependent steps of protein transport, the exact role of Tim44 is far less clear. Indeed the data presented in this manuscript would suggest a role of Tim44 also at an earlier stage of protein transport, which could affect both sorted as well as matrix-targeted precursors. Also, we have previously seen that Tim44 can be crosslinked to laterally sorted precursors (Popov-Celeketic et al., EMBO Reports, 2011), even when their transport does not depend on the ATPase activity of the import motor. However, as differential sorting of precursors by the TIM23 complex is not a focus of this manuscript, we removed this point from the text to simplify it for non-specialists.

We measured the membrane potential in FL and N+C mitochondria and, not surprisingly, observed its slight reduction in N+C mitochondria, in agreement with the very strong growth defect of N+C cells and with the established essential role of Tim44 in transport of a number of components of respiratory chain complexes and their assembly factors into mitochondria. In this respect, it is also worth noting that even in glucose medium at either 24°C or 30°C, N+C cells grow about three times slower than FL. In addition, rather than analyzing import of AAC by another method, we analyzed import of another TIM23-independent substrate, Tim23 itself. Like with AAC, we also observed no major difference in its import efficiency in FL and N+C mitochondria. These data are now included in Figure 3 and in the Results section.

*3) Ting et al. (2014) reported that the N-terminal domain of Tim44 interacts with Tim23 as well as motor components by using crosslinking in intact mitochondria. Using purified Tim44 domains and mitochondria lysed with detergent, the authors confirmed the interaction of the N-terminal domain with motor components, however, not with Tim23. They speculate that crosslinking of the N-terminal Tim44 domain to Tim23 requires a specific conformation of Tim23 only adopted when Tim23 is bound to Tim17 in the inner membrane (i.e. in intact mitochondria). Since the authors demonstrate the interaction of the C-terminal domain of Tim44 with Tim17 by using the purified domains and lysed mitochondria, but use crosslinking in intact mitochondria for showing the interaction with precursor proteins, the two approaches should be compared more directly. A suitable approach would be to analyze the interaction of the C-terminal domain of Tim44 with Tim17 in intact mitochondria by crosslinking. They may use radiolabeled Tim17 imported into mitochondria and their domain-specific Tim44 antibodies to address this topic.*

We agree that it would be excellent to demonstrate a direct Tim17-Tim44 interaction also by another assay such as crosslinking. However, to the best of our knowledge, no crosslinks of Tim17 to any other component of the TIM23 complex have ever been seen in wild type mitochondria, radiolabeled or not, not even to Tim23, even though their interaction was demonstrated by several different assays. The only crosslinks to Tim17 observed so far were upon a very comprehensive cysteine-scanning mutagenesis of Tim23 followed by cysteine-specific crosslinking (Alder et al., MBC, 2008) and the above mentioned paper by Ting et al. that used site-specific UV crosslinking to demonstrate an interaction between Tim17 and Pam17. Since the surfaces of either Tim17 or Tim44 involved in this interaction remain currently unknown, we feel that undertaking such a comprehensive site-specific approach would be beyond the scope of this manuscript.

*4) The purification experiments shown in Figure 6 and Figure 4 will benefit from quantification/statistics. In case of Figure 6, the outcome is clearly visible in the presented experiment, but this seems to be only a representative (of how many experiments?). The statistical analysis of repetitions would provide more confidence for this important conclusion. In the case of Figure 4, the conclusion about the presence of the N-and C-domains is clear looking at the presented experiment (representative?), in contrast to the conclusion concerning Tim21 and Pam17. If the authors would like to keep the observation concerning Tim21 and Pam17 in the text, it requires statistics.*

*Figure 4: It is difficult to compare the co-immunoprecipitation efficiencies of different TIM23 components since the membranes have been strongly trimmed and the blots are overexposed. Here different exposures might have to be presented.* All the experiments presented in this manuscript are representatives of experiments repeated at least three times. In addition, all the critical experiments were not only performed with at least three biological replicates but were also repeated by at least two different researchers to make sure that all the observed effects are reproducible.

Following the suggestion of the reviewers, we included a panel containing quantifications of binding of mtHsp70, as a representative of the import motor components, and of Tim17 to the various beads that were obtained in three different experiments. In addition, we added to the Figure legend the information that input contains 4.5% of the material used for binding. This information was unfortunately missing in the original submission.

The original Figure 4 was trimmed in order to remove the supernatants after coimmunoprecipitations. We felt this would make the figure easier for non-specialists. However, this certainly made it more difficult to judge the efficiencies of different coimmunoprecipitations. The new Figure 4 now includes the supernatants, so that the efficiencies can be directly compared. We agree with the reviewers on the point that the observations with Tim21 and Pam17, though certainly reproducible, as explained above, are not essential for the conclusions of the manuscript. We have therefore removed them both from the Figure and from the text, to make it clearer for non-specialists. In addition, the Figure legend now contains the information that the total and supernatant fractions contain 20% of the material used for coimmunoprecipitations. As above, we unfortunately forgot to add this information during original submission.

*5) Figure 4: There is far less Tim44 N+C associated with the TIM23 complex than full-length Tim44 in WT. This raises the question to what extend a Tim44-depletion mirrors the observed effects for the split protein with regard to crosslinks that do not include Tim44 (Tim23-Tim23, Tim16-Tim16) or import defects of sorted proteins.*

The new Figure 4, which now contains the supernatants after coimmunoprecipitations, also addresses this question. In addition, we used for western blot Tim44 antibodies prepared by mixing affinity-purified antibodies to N and C domains so that both domains are now roughly equally well visible. We would also like to note that the Tim44 signal in FL is expected to be stronger as antibodies to both domains recognize the full-length protein whereas the individual domains are recognized only by certain antibodies. We think that from the new Figure 4 it is clear that there is no less Tim44 bound to TIM23 in N+C than in FL. We would also like to add that it was previously shown that, upon depletion of Tim44 from cells, both Tim17-Tim23 and Tim14-Tim16 subcomplexes remain intact but they are completely dissociated from each other (Kozany et al., NSMB, 2004). In our opinion, it is clear from the presented coimmunoprecipitations experiment that the two subcomplexes remain associated in N+C mitochondria, unlike in cells depleted of Tim44.

6) The conclusion that the TIM23 complex adopts an "altered conformation" in N+C mitochondria is based on crosslinking experiments. It would be good to support this conclusion by an independent approach, such as native gel electrophoresis or differential protease sensitivity of the TIM23 components.

*Moreover, the term "altered conformational flexibility" is not fully fitting to the results presented here. The authors rather look at an altered subunit composition and not protein conformations. The text should be changed.*

We greatly appreciate this suggestion. We analyzed the TIM23 complex by blue native PAGE and indeed observed a changed running behavior of both Tim17 and Tim23 in N+C mitochondria as compared to FL. Since this alteration mainly affected the 90 kDa complex whose only known components are Tim17 and Tim23, this finding provides further support for our notion that the conformation, rather than the composition, of the TIM23 complex is affected in N+C mitochondria. This new finding is now included in Figure 5 and is described in the text.

As also discussed above, since the coimmunoprecipitation experiments revealed no major changes in the composition of the complex and both crosslinking and BN-PAGE support the notion of the changed conformation of the complex, we think that the term “altered conformational flexibility” is fitting our results and is supported by them.

*7) The part on the human patient mutation mimicked in yeast Tim44 is interesting, but not required for the rest of the story. Does the mutation in Tim44 affect the TIM23 complex or protein import in yeast or humans? As it stands it is not very informative. The authors should either remove that part or perform a functional analysis, i.e. test mutant cells/mitochondria for the functionality of the Tim44 C-terminal domain in the binding of Tim17 and precursor binding during mitochondrial import.*

We agree that the patient mutation mimicked in yeast Tim44 is not required to understand the rest of the story and have therefore removed this finding from the Abstract. As we also previously stated in the Discussion, we have performed initial analyses of yeast cells carrying this mutation and we observed neither an obvious growth nor an obvious in vivo import defect in these cells. Both these observations are in agreement with the relatively mild effect seen in the affected patients (as discussed in the text, mutations that would dramatically reduce the activity of the TIM23 complex would likely be embryonically lethal in humans). We did however observe that the mutation partially destabilizes the protein and we feel that this is an important observation as it may provide a clue as to what goes wrong in the affected patients and may stimulate studies with patient cell lines, which are clearly beyond the scope of this manuscript. We would thus prefer to keep this finding in the manuscript but are happy to leave the final decision to the editors.